# Two Decades of Evolution of Our Understanding of the Transient Receptor Potential Melastatin 2 (TRPM2) Cation Channel

**DOI:** 10.3390/life11050397

**Published:** 2021-04-27

**Authors:** Andras Szollosi

**Affiliations:** 1Department of Medical Biochemistry, Semmelweis University, 1085 Budapest, Hungary; szollosi.andras@med.semmelweis-univ.hu; 2MTA-SE Lendület Ion Channel Research Group, Semmelweis University, 1085 Budapest, Hungary; 3HCEMM-SE Molecular Channelopathies Research Group, Semmelweis University, 1085 Budapest, Hungary

**Keywords:** TRPM2, ion channels, single particle cryo-EM, ADP-ribose, Nudix hydrolase

## Abstract

The transient receptor potential melastatin (TRPM) family belongs to the superfamily of TRP ion channels. It consists of eight family members that are involved in a plethora of cellular functions. TRPM2 is a homotetrameric Ca^2+^-permeable cation channel activated upon oxidative stress and is important, among others, for body heat control, immune cell activation and insulin secretion. Invertebrate TRPM2 proteins are channel enzymes; they hydrolyze the activating ligand, ADP-ribose, which is likely important for functional regulation. Since its cloning in 1998, the understanding of the biophysical properties of the channel has greatly advanced due to a vast number of structure–function studies. The physiological regulators of the channel have been identified and characterized in cell-free systems. In the wake of the recent structural biochemistry revolution, several TRPM2 cryo-EM structures have been published. These structures have helped to understand the general features of the channel, but at the same time have revealed unexplained mechanistic differences among channel orthologues. The present review aims at depicting the major research lines in TRPM2 structure-function. It discusses biophysical properties of the pore and the mode of action of direct channel effectors, and interprets these functional properties on the basis of recent three-dimensional structural models.

## 1. Introduction

The transient receptor potential (TRP) superfamily is a large cluster of transmembrane proteins consisting of ion channels that are expressed in a large variety of tissues. The superfamily falls into several families: canonical TRPC (1–7), vanilloid TRPV (1–6), vanilloid-like TRPVL, melastatin-like TRPM (1–8), polycystin TRPP (2, 3 and 5), mucolipin TRPML (1–3), ankyrin-rich TRPA (7 subfamilies), soromelastatin TRPS and no mechanoreceptor potential C TRPN (or NOMPC) [1]. TRP channels sense a plethora of signals, including ion concentration changes in the cell, binding of lipids or other ligands, temperature changes and mechanical signals, and respond with pore opening and transmembrane cation flux [2]. The largest family is TRPM, which has eight members. Most TRPM channels conduct both mono- and divalent cations, while TRPM4 and 5 are Ca^2+^-impermeable [3,4].

This review focuses on structure–function aspects of TRPM2 channels, which are widely expressed in bone marrow, the heart, liver, pancreas, leukocytes, lungs, spleen, eye, and brain [5,6,7]. TRPM2 is a nonselective cation channel which is co-activated by ADP-ribose (ADPR) and Ca^2+^. TRPM2 activity is indispensable for many physiological processes including insulin secretion in pancreatic β-cells [8], monocyte chemokine production [9] and heat sensation of hypothalamic neurons [10]. Due to its permeability to Ca^2+^, TRPM2 is also involved in many pathophysiological processes that lead to cell death [11], including neuronal cell death upon reactive oxygen species (ROS) production and in neurodegenerative disorders [5,12,13,14]. Hence, TRPM2 has become an attractive pharmacological target.

TRPM2 is a homotetramer; its subunits are built from ~1500 amino acid residues and consist of a large cytosolic N-terminal region, followed by a transmembrane domain (TMD) and a cytosolic C-terminal region. Both the intracellular and transmembrane regions are involved in ligand binding. The TMD contains six transmembrane helices (S1–6), and its structural organization resembles that of voltage-dependent cation channels [15,16]. In voltage-gated K^+^ channels S1–S4 constitute the voltage sensor domain (VSD) which harbors four conserved arginines that move in response to voltage changes (the “gating charge”), causing strong voltage sensitivity of channel gating. In the TRPM family, the overall structural organization of the VSD is conserved, but the four arginines are absent. Correspondingly, TRPM4, 5 and 8 are weakly voltage sensitive, while TRPM2 is largely voltage insensitive. The S5–S6 region forms the pore domain (PD) which embraces an extracellular non-selective cation permissive selectivity filter and a cytoplasmic gate that regulates channel open probability. TRPM2 is unique among other family members in that its C-terminal region is extended by an ~270 amino acid-long cytosolic domain which shares ~40% sequence homology with the mitochondrial NUDT9 protein [7] and is hence called NUDT9-homology (NUDT9-H) domain. The mitochondrial NUDT9 enzyme belongs to the group of nucleoside diphosphate-linked moiety X (Nudix) hydrolases, which all contain a canonical Nudix motif sequence, GX_5_EX_7_REUXEEXGU, where X refers to any, while U to a large hydrophobic residue [17]. The Nudix motif, and especially the Nudix box (REUXEE), is thought to be important for the stability of the active site and coordination of cationic cofactors. Mutations introduced into canonical Nudix-box positions abolish hydrolytic activity of Nudix hydrolases, as shown, e.g., for NUDT9 by RILGEE and REFGKK substitutions of its Nudix box (REFGEE) [18]. The mechanism of the enzymatic reaction is general base catalysis. In pyrophosphatases, such as NUDT9, a water-driven nucleophilic attack on the alpha phosphorus initiates hydrolysis and is facilitated by divalent cation cofactors, typically Mg^2+^. The nature of the catalytic base is variable, either a glutamate within the Nudix box [19] or a glutamate [20] or histidine outside the Nudix sequence [21]. NUDT9 is a specific ADP-ribose (ADPR) pyrophosphatase (ADPRase), which cleaves ADPR into AMP and ribose-5-phosphate. An N-terminal subdomain (cap), consisting of β-hairpins and α-helices, caps the C-terminal subdomain (core). The core subdomain, which consists of several α-helices and pleated β-sheets, harbors the catalytic site, and shows the highest homology with other Nudix hydrolases [22]. While the isolated core subdomain retains activity and specificity for ADPR, the cap subdomain is also involved in correct substrate positioning [18]. ADPR supposedly binds into a cleft between the core and the cap subdomain, where its hydrolysis product ribose-5-phosphate was detected in structures [22].

In cells, ADPR is produced from several metabolic pathways. Sir2 (class III protein deacetylase) enzymes catalyze the removal of lysine-bound acetyl groups and use NAD as a cofactor. One reaction end product is 2’- or 3’-O-acetyl-ADPR (OAADPR), which is converted into ADPR by several cytosolic enzymes [23]. Interestingly, OAADPR itself was reported to bind to the NUDT9-H domain of human TRPM2 (hsTRPM2) with a similar affinity as ADPR [24], while mitochondrial NUDT9 binds OAADPR with a 500-fold lower affinity than ADPR. ROS generation in either immune cells or in the brain upon ischemic stroke elevates cytosolic ADPR, leading to Ca^2+^ entry into cells through TRPM2 channels. ROS, on the one hand, induce the activation of the DNA repair enzymes poly-ADPR polymerase and glycohydrolase, which produce cytosolic ADPR. On the other hand, the ADPR precursor NAD^+^ and possibly ADPR itself might be released from the mitochondrial matrix through the mitochondrial permeability transition pore [11].

## 2. Characterization of TRPM2 in a Cell-Free System and Identification of Direct Effectors

### 2.1. Activation of TRPM2 by ADPR, Ca^2+^ and Phosphatidylinositol-4,5-bisphosphate (PIP_2_)

Soon after cloning of the TRPM2 channel [25], whole-cell patch-clamp studies showed that the channel is activated by intracellular ADPR and activation requires Ca^2+^ [7,26,27,28,29]. As in whole-cell studies the composition of the cytosolic solution, and particularly that of microdomains, is not strictly controlled, the location of the activating Ca^2+^ binding sites (extra- or intracellular) could not be unambiguously determined [27,29]. In contrast, inside-out patch clamp recordings afford strict control over the composition of the cytosolic solution, and rapid application/removal (on the timescale of tens of milliseconds) of intracellular ligands. In patches excised from *Xenopus laevis* oocytes injected with hsTRPM2 cRNA, large TRPM2 currents were evoked by simultaneous application of ADPR and Ca^2+^ (Figure 1A) [30,31]. Both ligands were required for channel activity, but interestingly, effective concentrations differed from those published earlier in studies using whole cells [7,24,26,27,28,29]. Whereas in whole-cell recordings EC_50_ for ADPR and Ca^2+^ were ~100 μM and ~300 nM, respectively [7,27], in inside-out patches, ADPR apparent affinity was two orders of magnitude higher (EC_50_~1 μM) whilst that of Ca^2+^ was two orders of magnitude lower (EC_50_~20 μM), and these apparent affinities were little affected by the concentration of the other ligand [30,31]. A possible explanation for a lower Ca^2+^ affinity could have been the loss of calmodulin in the patch (cf [29,32]); however, the addition of external bovine calmodulin had no effect on fractional currents at micromolar Ca^2+^ concentrations [31]. Since even 1 mM ADPR elicited only minimal activation in submicromolar Ca^2+^, the TRPM2 channel seems to be intrinsically set to sense high local Ca^2+^ concentrations.

While no structural data were available at the time, functional observations in inside-out patches allowed rough localization of the Ca^2+^ binding sites. In steady-state single channel recordings, channel closed-time durations were independent of extracellular Ca^2+^ concentration, suggesting that in closed channels the binding sites are shielded from extracellular Ca^2+^ [30]. On the other hand, macroscopic channel closure upon sudden intracellular Ca^2+^ withdrawal could be incrementally slowed by raising extracellular Ca^2+^ [30], suggesting that as long as the pore is open extracellular Ca^2+^ can reach its binding sites despite Ca^2+^-free vectorial rinsing of the cytosolic channel surface. These findings strongly suggested that the Ca^2+^ binding sites are located in an intracellular crevice very near the channel pore.

Besides the two obligate ligands, ADPR and Ca^2+^, the presence of membrane phosphatidylinositol-4,5-bisphosphate (PIP_2_) is also a prerequisite for channel activation, as in inside-out patches masking the negatively charged headgroups of endogenous PIP_2_ with the polycation polylysine shuts TRPM2 pores, which can then be re-opened only by millimolar amounts of Ca^2+^ [36] or by administration with external PIP_2_ [33]. While H_2_O_2_ potently stimulates TRPM2 currents in intact cells [5,26,37], this effect must be a secondary consequence of ADPR and/or Ca^2+^ release in response to the induced oxidative stress [37], as H_2_O_2_ itself was ruled out as a direct effector of TRPM2 channel activity in cell-free recordings [31].

### 2.2. Pyridine Nucleotides and Their Derivatives Are Not Direct Activators of TRPM2

In the body, CD38, a multifunctional glycoprotein enzyme present on the surface of immune cells, converts the pyridine nucleotides NAD and NAAD into ADPR, but NADP and NAADP into ADPR-2-phosphate (ADPRP) [38]. These nucleotides and others, the metabolism of which is intertwined with that of ADPR, were tested to address the potential effects on TRPM2. NAD was identified as a channel agonist in early studies [5,28], while no binding to the NUDT9-H domain was later reported [24]. NAAD [31], NAADP [31,39,40] and cADPR [39,40] were reported to act as activators, AMP as an inhibitor [24,26], while NADH and NADP were found to be ineffective [5]. Besides these reported effects on channel activity, cADPR [26,39,40] and NAADP [39,40] were also shown to augment the effects of ADPR in a positively cooperative manner. 8-Br-cADPR, an agent shown to reduce ischæmic acute kidney injury [39], was suggested to act as a partial TRPM2 agonist and a competitive inhibitor of activation by cADPR [26]. Most of the above studies examined the effects of nucleotides in whole-cell systems. In a study in inside-out patches, the effects of cADPR on TRPM2 channel activity were shown to be attributable to ADPR contamination [31]. Decontamination of commercial cADPR stocks with a nucleotide pyrophosphatase which selectively cleaves ADPR but not cADPR eliminated channel stimulation by the compound in excised patches. Furthermore, no cooperative effects of cADPR with ADPR were observed. Recently, Yu et al reported that synthetic pure cADPR directly binds to the human NUDT9-H domain [41]. SPR titration indicated an apparent affinity (K_d_) of ~12 μM; however, much higher concentrations of cADPR (EC_50_ ~250 μM) were required to stimulate macroscopic TRPM2 currents. Considering its submicromolar cellular concentration [42], cADPR does not seem to act as a physiological activator of TRPM2. TRPM2 channel activation by various pyridine dinucleotides is explained by their spontaneous degradation, which also produces ADPR(P). Cleavage of contaminating ADPR in NAD and NAAD stocks, or of contaminating ADPRP in NAADP stocks, by the purified NUDT9 ADPRase abolished the stimulatory effects of the dinucleotides, ruling them out as direct effectors of TRPM2 [43]. Moreover, these experiments also demonstrated that none of the spontaneous and/or enzymatic cleavage products—nicotinamide, nicotinic acid, AMP(P) or ribose-5-phosphate—are TRPM2 channel activators. An apparent stimulatory effect of NAADP prior to decontamination pinpointed its spontaneous degradation product ADPRP as an agonist. Indeed, in inside-out patches, pure ADPRP opened TRPM2 channels, albeit its apparent affinity was lower (EC_50_~13 μM) and maximal stimulation smaller (~80%) when compared to ADPR. The lower open probability supported by ADPRP was explained by an approximately three-times faster macroscopic closing rate, reflecting a shortened open burst [43]. A similar efficacy but even lower affinity (EC_50_~110 μM) for ADPRP was reported in a whole-cell study [44] that also showed 2’-deoxy-ADPR (dADPR) to act as a superagonist on hsTRPM2 channels. In the presence of dADPR, whole-cell TRPM2 currents were 37% larger, and following patch excision channels inactivated slower, but the EC_50_ of dADPR was similar to that of ADPR. Since upon oxidative stress-induced DNA damage, dADPR may accumulate due to CD38 activity, this nucleotide is a likely regulator of TRPM2 function under pathophysiological conditions.

In summary, recordings in cell-free systems demonstrated that (d)ADPR is a physiologically relevant activator of the hsTRPM2 channel and ADPRP is a partial agonist, whereas AMP and pyridine dinucleotides are without effect [31,43] and cADPR is probably a non-physiological low affinity agonist [41]. Note, however, that nucleotide affinity and efficacy profiles differ among channel orthologues (discussed in a later chapter). Discussion of pharmacologically designed analogs of ADPR, with adducts on either the adenine base or on the terminal ribose, are beyond the scope of the current review, although some of them might be exploited as therapeutic drugs [6,45].

## 3. Gating Properties of the TRPM2 Channel

### 3.1. hsTRPM2 Is Not an Active ADPR Hydrolase

The early finding that hsTRPM2 binds and hydrolyses ADPR [7] classified TRPM2 among channel-enzymes (“chanzymes”), a very underrepresented class of ion channels. To date, the only other known members are CFTR, an ATP-binding cassette-type anion channel; and TRPM6 and 7, which have cytosolic kinase domains. While CFTR obeys a non-equilibrium cyclic gating mechanism in which ATP binding at nucleotide binding domains opens and ligand hydrolysis closes the channel pore [46,47], kinase activity of TRPM6/7 channel cytosolic domains is not coupled to gating conformational changes of their pores [48].

While identification of a novel chanzyme was an intriguing novelty, ADPRase activity of NUDT9-H was debated in the literature. On the one hand, the isolated NUDT9-H domain of hsTRPM2 was reported to operate as an active ADPR hydrolase with substrate preference towards ADPR over ATP, albeit with a maximum catalytic activity (~0.1 s^−1^) approximately ~100-fold lower than that of mitochondrial NUDT9 (~10 s^−1^) [7,18]. Differences in hydrolytic efficiency were linked to non-canonical substitutions (REF-to-RIL) in the Nudix box region of the NUDT9-H domain, since the analogous double mutation similarly reduced NUDT9 activity (by ~100-fold). As a double lysine substitution of two glutamates in the Nudix sequence, deleterious for NUDT9 ADPRase activity [18], did not have a major impact on whole-cell TRPM2 current amplitudes [37], it was concluded that ADPR hydrolase activity is not required for channel gating. At the same time, abrogation of hsTRPM2 channel currents in intact cells, even at extremely high ligand concentrations, by the RIL-to-REF mutation [37,49] was interpreted to reflect increased ADPRase activity of the NUDT9-H domain and consequent rapid clearance of the activating substrate [49]. On the other hand, another study detected no measurable ADPRase activity for the purified NUDT9-H domain [24], although calorimetric titration reported a binding affinity for ADPR (~100 uM) very similar to that observed in previous studies [7,18]. The reason for this discrepancy seemed unclear, as all three studies used the same sensitive colorimetric assay, which detects inorganic phosphate released by alkaline phosphatase from the ADPR cleavage products [50,51]. The conflicting results could have originated from different protein purification and enzymatic assay conditions, since the expression conditions were rather similar. Tagless NUDT9-H protein was purified by salting out and subsequent ion exchange and ADPR hydrolysis assay was performed at basic pH in one group [7,18], while the other laboratory purified a C-terminally His_6_-tagged construct by affinity chromatography and measured enzyme activity at close to neutral pH [24].

Due to these contradictory reports, two important questions remained open at the time: (1) Does the NUDT9-H possess intrinsic hydrolytic activity? (2) Is the putative slow ADPR hydrolysis cycle at the NUDT9-H domain coupled to the similarly slow gating transitions of the hsTRPM2 pore, or does hydrolysis occur independently from channel gating?

Unfortunately, whereas ADPR- and Ca^2+^-activated whole-cell currents of hsTRPM2 expressed in mammalian cell lines persist for up to 20 min without marked decay [7], in inside-out patches, a rapid loss of hsTRPM2 channel activity (“rundown”) (Figure 1B, left) [30] makes the analysis of gating problematic. The rundown reflects a fast permanent inactivation of channels in the patch rather than a gradual decline in channel open probability [30]. It is likely caused by a conformational change in the outer pore region, as a GY-to-LDE substitution that introduces two negatively charged residues near the selectivity filter and renders the pore sequence similar to that of TRPM5 (Figure 2, green box) prevents inactivation [36]. Such “TRPM5-like” (T5L) hsTRPM2 channels thus allow for longer recordings while their gating kinetics otherwise resemble that of WT hsTRPM2. Thus, this construct proved to be a convenient model channel for more accurate analysis of hsTRPM2 gating properties. Mutations that are expected to interfere with the binding and hydrolysis of ADPR at NUDT9-H had no effect on the macroscopic gating properties of T5L TRPM2, strongly suggesting that ADPR hydrolysis is not linked to conformational transitions of the channel [52]. Further supporting evidence for such a lack of coupling came from experiments using a non-hydrolysable analog of ADPR. In α-β-methylene-ADPR (AMPCPR), the two phosphates are bridged by a methylene group [53], rendering it resistant to hydrolysis by NUDT9 [52]. Nevertheless, AMPCPR readily opened T5L hsTRPM2 channels, although with a ~50-fold lower apparent affinity and ~2-fold lower maximal efficacy when compared to ADPR. Macroscopic current decay upon nucleotide removal was not slower for TRPM2 currents evoked by AMPCPR compared to those evoked by ADPR, which would have been expected if ligand hydrolysis terminated open channel bursts. Rather, the closing rate of channels opened by AMPCPR seemed to be slightly accelerated. Taken together, these experiments ruled out ADPR hydrolysis to be coupled to either pore opening or pore closure, and implied that gating of hsTRPM2 is at thermodynamic equilibrium, as for conventional ligand-gated channels.

Enzymatic assays on the purified human NUDT9-H protein (hsNUDT9-H) are largely hampered by its low solubility [35]. This problem could be overcome by constructing a series of chimeras from complementary segments of hsNUDT9-H and mitochondrial NUDT9. A chimera in which only a short C-terminal sequence of hsNUDT9-H was replaced by that of NUDT9, with 90% of its sequence identical to that of hsNUDT9-H, was a soluble and monodisperse protein and showed no signs of ADPRase activity. Moreover, all chimeric constructs that contained the Nudix box sequence of hsNUDT9-H were catalytically incompetent, whereas chimeras comprising the canonical NUDT9 Nudix box all hydrolyzed ADPR independent of their C-terminal sequences. Interestingly, a slow accumulation of AMP and ribose-5-phosphate was detectable even without added protein in reaction mixes at alkaline pH. As a reference, at pH = 9, almost 1% of ADPR spontaneously degraded within half an hour at 37 °C, yielding AMP and ribose-5-phosphate in sufficient amounts to be detected by the used colorimetric assay [35]. The rate of such slow spontaneous ADPR degradation at basic pH, which is not observed at neutral pH, fully explains the previously reported slow NUDT9-H activity [7,18]. Taken together, ADPR is a primary activator of hsTRPM2 channels; it binds to hsTRPM2 and activates channels with an apparent affinity of 1 to 2 μM; however, the ligand is not hydrolyzed by the hsNUDT9-H domain.

### 3.2. Invertebrate TRPM2 Channels Are Active ADPRase Chanzymes

Careful comparison of NUDT9-H sequences among animal phyla (Figure 1C, right) revealed canonical Nudix regions in invertebrates (Figure 1C, right, blue sequences), but the evolutionary appearance of several non-conserved substitutions at key glutamate residues (Figure 1C, right, asterisks) in vertebrates [34] (Figure 1C, right, red sequences). The human TRPM2 channel was shown to be enzymatically inactive (Figure 1D, center) [35], while a more ancient TRPM2 orthologue from the cnidarian *Nematostella vectensis* (nvTRPM2) was suggested to possess ADPRase activity [56]. This raised the possibility of a general evolutionary trend for invertebrate channels to function as active ADPR hydrolyze chanzymes. That idea was tested on purified full length TRPM2 proteins and isolated NUDT9-H domains of two ancient orthologues [34]. These proteins all proved catalytically active: in the presence of Mg^2+^ and alkaline pH nvTRPM2 hydrolyzed ADPR (Figure 1D, right) with a turnover number of ~40 s^−1^ per subunit (Figure 1E), while TRPM2 from the unicellular choanoflagellate *Salpingoeca rosetta* (srTRPM2) was a slower chanzyme (k_cat_ of ~2 s^−1^ per subunit). The high sensitivity of the assay [51] allowed both activities to be clearly distinguished from spontaneous ADPR cleavage.

The molecular turnover rate of nvTRPM2 was dependent on free Mg^2+^ concentration and the relationship was best fit with a Hill-function with a coefficient of ~2, indicating the role of two Mg^2+^ ions in ligand stabilization at the active site [34]. While this coefficient was little affected, maximal catalytic activity was progressively reduced and the apparent affinity decreased at lower pH values. In contrast, H^+^ concentration dependence of the catalytic constant was described by a Michaelis–Menten function, suggesting that a single protonatable group plays a central role in substrate hydrolysis; the pK_a_ of this residue was sensitive to Mg^2+^ concentration. The mutual influence of H^+^ and Mg^2+^ concentration on the apparent affinity of the other ion suggests that the key protonatable residue, which likely serves as the catalytic base, is near the bound Mg^2+^ ions, a prediction that fits well with the mechanism proposed for several other Nudix hydrolases [17]. The Mg^2+^- and pH-dependence of srTRPM2 channels was more complex; Mg^2+^-dependence of the molecular turnover number was best fit by a double Hill-function, one of which was sensitive to pH [34]. The data could be fitted to a model in which catalysis is supported by the binding of ~2 Mg^2+^ ions to a high-affinity site near the catalytically important protonatable residue, while at tens of millimolar Mg^2+^ concentrations, a third Mg^2+^ ion further augments ADPR hydrolase activity by binding to a low-affinity site more distant from the catalytic group. While the potential functional relevance of such species-specific differences in catalytic mechanisms needs to be established, these findings support the prediction that all invertebrate TRPM2 proteins are chanzymes.

When compared to typical values of 10^4^–10^6^ s^−1^ of substrate diffusion-limited enzymes, the catalytic constants of nvTRPM2 (Figure 1E) and srTRPM2 are slow, as are the rates of gating transitions of TRPM2 channels. Theoretically, in invertebrate channels, some degree of coupling might exist between the ligand hydrolysis cycle at the NUDT9-H domains and gating of the channel pore. In that case, the prevention of ADPR hydrolysis would be expected to cause some change in gating pattern. That possibility was ruled out by experiments that revealed no major changes in nvTRPM2 channel gating upon disruption of ADPR hydrolysis by NUDT9-H domain catalytic site mutations, complete removal of Mg^2+^, or the use of the non-hydrolyzable ADPR analog AMPCPR [34]. Thus, the ADPR hydrolase activity of invertebrate NUDT9-H domains is not coupled to pore gating and must have evolved to serve some other role, perhaps clearance of the activating ligand from the cytosol. Lastly, even complete deletion of the NUDT9-H domain failed to abolish nvTRPM2 channel currents [56], and did not substantially alter channel gating properties [57]. Besides supporting the above conclusion, this shocking finding also introduced a further level of complexity by strongly indicating the presence of a second ADPR binding site within the structure.

## 4. Structure of TRPM2 Solved by Single Particle Cryo-EM

Just as the picture of TRPM2 channel gating seemed to converge towards a consensus view, reversible non-hydrolytic binding/dissociation of ADPR at the NUDT9-H domain driving channel gating, a set of major surprises emerged. Whereas truncation or complete deletion of the catalytically inactive NUDT9-H domain in human and zebrafish (*Danio rerio*) TRPM2 channels was known to abolish whole cell currents [37,49,58,59], a similar deletion of the catalytically active NUDT9-H domain in nvTRPM2 had little impact on gating [56,57]. This finding strongly suggested the existence of a second ADPR binding site in invertebrate channels, which is solely responsible for channel activation. However, where is this second site located, and does it also exist in vertebrate TRPM2 proteins? Through which site does ADPR regulate vertebrate TRPM2 channel activity, and what is the relevance of dual or sequential substrate binding? These challenging questions further augmented the need for high resolution TRPM2 structures.

### 4.1. The Resolution Revolution of Single-Particle Cryo-EM

To support the aim of the special issue to confer current approaches, the cryo-electron microscopy (cryo-EM) technique is briefly discussed here. In 2017, the Nobel prize in Chemistry was awarded to Jacques Dubochet, Joachim Frank and Richard Henderson for their continuous efforts to develop the method. The general pipeline of the technique is as follows. Tiny volumes of purified and concentrated proteins are loaded onto microscope grids. Rapidly frozen grids are placed into cold stages of an electron microscope and the specimen is hit by a high-energy electron beam, which generates a heavily magnified 2D projection image of each of the specimen particles at the image plane of the detector. Projection images appear as contrasting silhouettes of the original particles. Since particles are spread in the sample in random orientations, they are imaged from a vast number of different angles. Projections of similarly oriented particles can be clustered and averaged, thereby greatly increasing spatial resolution. Finally, the 3D shape (electron density) of the particles can be computationally reconstructed from the ensemble of the 2D projections. As the technique does not require cumbersome crystallization of the target protein, it has lately become the golden standard for protein structure determination, yielding >2000 deposited structures last year. However, it had to come a long way to become a “routine” technique.

In the 1970s, X-ray crystallography dominated the field of protein structural biology. Electron microscopy was employed only for low-resolution analysis of larger particles, such as dead material, typically embedded in a polymer. Pioneers of cryo-EM hence evolved from crystallographers, and at first preferentially studied electron diffraction of 2D and 3D crystals. In 1975, Henderson’s lab published a 7 Å resolution 3D electron microscopy structure of bacteriorhodopsin obtained from 2D crystals without negative stain [60]. It took more than 10 years of effort to improve cryo-EM from such low-resolution rendering of 3D protein shape (“blobology”) into near-atomic (~3.5 Å [61]) resolution reconstruction of protein electron density maps. During this time, several challenges had to be overcome, marking milestones in the history of cryo-EM.

One long-standing problem in electron microscopy was radiation damage caused by exposure to high energy electrons, which induces ionization and chemical remodeling of the sample. The freezing of thin 3D crystals greatly improved the quality of the collected images by reducing radiation damage [62]. The rapid cooling of samples was perfected by the Dubochet lab. A droplet of sample is blotted onto a tiny microscope grid, which is then plunge-frozen in liquid ethane cooled to the temperature of liquid nitrogen [63]. The process leads to the formation of a thin film of the sample, captured in vitreous ice. Such a frozen but aqueous sample is more resistant both to radiation damage and to dehydration caused by the strong vacuum required for a spatially and temporally coherent electron beam in the microscope.

Joachim Frank’s group introduced a statistical analysis method to extract class averages of 2D particle projections in noisy images [64]. Clustering of similarly oriented particles greatly improved resolution. In many cases, the sample is irradiated with an electron beam perpendicular to its plane. Selected classes of 2D particle projections are aligned and a 3D density is computed; however, if certain orientations are underrepresented, this will affect the quality of the initial 3D model. Another approach uses tilt pairs of images and often generates more detailed 3D reconstructions. In the latter process, the same set of particles is imaged from two different angles; hence, each particle will be detected in two views that differ by a known angle. The initial 3D model can be computed from the ensemble of such “tilt pairs” of particle classes. The combination of plunge-freezing of samples, the use of tilt pairs, and 2D classification practically established single-particle cryo-EM and led to the publication of the first structural reconstruction of the ribosome [65].

One major area of improvement was the technology of various parts of the microscope itself. The application of higher vacuum, more coherent and higher energy (typically 300 keV) electron beams, and more advanced cold stages all improved image sampling. Up until 2012, micrographs were recorded with scintillation devices on photographic film and were later digitized. A major breakthrough in improving image quality arrived with the use of direct electron detectors. Exquisite thinness of the sensory layer of such cameras prevents blooming of the detected electron signal into neighboring detector zones, allowing for a very small physical pixel size (a few μm in width). Furthermore, such cameras also allow for high frame rate recordings, important for addressing the problem of beam-induced sample motion. The latter occurs as the ice layer melts and bulges upon irradiation, causing particles to move during detection, leading to image blurring. Several software packages were developed for beam-induced motion correction, all of which depend on tracking the movement of individual particles between sequentially obtained image frames. Due to smart software packages nowadays, most parts of the pipeline including image collection and analysis is (semi-)automated, rendering single-particle cryo-EM a particularly robust and highly user-friendly structure determination method.

### 4.2. General Assembly of the TRPM2 Channel

Up to now, 14 TRPM2 channel structures have been published, starting with the structure of nvTRPM2 (6co7) [33] which still has the highest resolution (3.1 Å) to date, followed shortly by structures of TRPM2 from *Danio rerio* (drTRPM2) (6drj, 6drk, 6d73, 6pkx, 6pkv, 6pkw) and hsTRPM2 (6mix, 6miz, 6mj2, 6puo, 6pur, 6pus, 6puu). The general assembly of all known TRPM2 channels resembles that of other TRPM-family proteins (Figure 3A). The bulk (>80%) of the homotetrameric protein resides in the cytosol (Figure 3C). A large, ~800 amino-acid, cytosolic N-terminal region and a shorter cytosolic C-terminal region flank the ~270 amino-acid transmembrane domain (TMD). Various nomenclatures have been adopted in the literature to describe TRPM cytosolic domains. The N-terminal part of the amino acid sequence contains four sequential “TRPM homology regions” (MHR1-4) shared among all family members (Figure 3B, shades of blue). In the structures, the MHR1/2 region forms a single domain (Figure 3B,C, dark blue), sometimes referred to as N-terminal domain (NTD) [33] or as nucleotide binding domain (NBD) in TRPM4 [66], in which inhibitory adenine nucleotides bind. The MHR3 region folds into a series of short helix–loop–helix motifs called ankyrin repeats, also found in other TRP channel subfamilies [67,68], and is therefore sometimes referred to as the ankyrin repeat domain (ARD, Figure 3B,C, cyan) [33,66]. The MHR4 region forms a large, mainly α-helical layer, the linker helical domain (LHD, Figure 3B,C, medium blue) [33,66], which connects the MHR1-3 region to the TMD. In the reports that describe the structures of hsTRPM2 [58,69] and drTRPM2 [59], the term MHR3 is used to denote the ARD plus the first two α-helices of the LHD, while the rest of the LHD is denoted as MHR4. For the sake of clarity, in the present review, the latter MHR terminology will be used for all TRPM2 orthologues. Thus, starting from the N-terminus, all TRPM-family channel protomers contain the following sequential polypeptide segments: MHR1/2, MHR3, MHR4, a short “pre-S1” helix (Figure 3B,C; orange) which attaches to the TMD, the S1–S4 segment (Figure 3B,C; yellow; in TRPM2, also called the voltage sensor-*like* domain (VSLD) for lack of voltage sensitivity); a short S4–S5 linker helix parallel to the cytosolic membrane surface, the S5–S6 segment (Figure 3B,C; green; also called the pore domain (PD)), the hallmark TRP helix (TRP H1; Figure 3B,C; red; an extension of the S6 helix that bends towards the inner membrane surface), a membrane-reentrant TRP helix 2 (TRP H2; Figure 3B,C; red), a subsequent TRP loop which connects to the long “stretcher” (or rib) helix (Figure 3B,C; pink), and finally, the coiled-coil region (Figure 3B,C; violet). The C-terminal NUDT9-H domain, unique to TRPM2, is observed only in the human and drTRPM2 channel structures [58,59,69] (Figure 3B,C, salmon). In the structures of all TRPM family channels, the conserved domains assemble into a three-tiered structure. MHR1/2/3 and the coiled-coil region form the square-shaped bottom tier (cf., Figure 3C). The tightly interacting MHR domains form a thick peripheral layer that surrounds a central cavity and is loosely connected to the vertical coiled coil segment traversing the latter (Figure 3D, top-right and bottom-left cross sections). The MHR4 domains also assemble into a square ring (Figure 3D; bottom-right cross section), which physically links the bottom to the top tier. The top tier is composed of the pre-S1 helices, the TMD, and TRP helices 1 and 2, and embraces the non-selective cation-permeable pore of TRPM2. In the hsTRPM2 and drTRPM2 structures, the NUDT9-H domains are seen to build a fourth tier below the bottom tier (Figure 3D; top-left cross section), and the latter two tiers form a ligand-sensing layer (Figure 3C). However, the precise interactions between the NUDT9-H domains and the MHR regions are variable among orthologues, which might underlie their different activation mechanisms, as will be discussed later. Key residues at the MHR1/2-MHR3 interface that form a high-affinity adenine nucleotide binding site in TRPM4 [66] are not conserved in TRPM2, explaining the lack of TRPM2 inhibition by ATP.

### 4.3. The TMD Region and Pore of the TRPM2 Channel

The MHR1-4 region is connected to the TMD through the pre-S1 helix, which, together with terminal MHR4 helices, forms an interaction network at the membrane-cytosol interface. The TMD region of TRPM2 closely resembles that of voltage-gated K^+^ channels. In the latter, 4–6 Arg and/or Lys residues per subunit in the S4 segment of the VSD bear the gating charges, which move towards the extracellular side upon membrane depolarization, causing pore opening [16]. A paddle-like dislocation of the S4 helices is translated through the S4–S5 linker into S6 movements that open the gate, a bundle formed by the cytosolic ends of the four S6 helices. In contrast, in TRPM2, the S4 segments hold only two conserved positively charged residues (and a non-conserved Arg at the very beginning of S4 in nvTRPM2; Figure 2, magenta boxes), and a tight hydrophobic interaction interface between S4 and S5 prevents major movements of the VSLD. Hence, the TRPM2 channel intrinsically has very little sensitivity to transmembrane potential changes. Similarly to K^+^ channels, the VSLD of TRPM2 is domain-swapped, i.e., the VLSD of a subunit connects to the PD of a neighboring subunit (Figure 3A, center), allowing the S4–S5 linker helix to be engaged in gate opening. The PD shows a K^+^ channel-like architecture [70] and is formed by the ascending S5 helix (Figure 4A, orange), an extracellular turret loop (Figure 4A, gray), the reentrant pore helix (Figure 4A, red), a short ascending selectivity filter loop (Figure 4A, green), a short post-filter helix (Figure 4A, yellow), and a post-filter loop (Figure 4A, cyan) which connects to the descending S6 helix (Figure 4A, blue). In a canonical K^+^ channel selectivity filter, the permeating ion is dehydrated, and is instead coordinated by carbonyl oxygens facing the permeation pathway. By mimicking the geometry of the hydration shell of a K^+^ ion, the filter ensures a high degree of selectivity for K^+^, and also diminishes the energy barrier for K^+^ transition through the pore, allowing a high permeation rate [71]. TRPM2 is a non-selective cation channel which, in contrast to its close relative TRPM4 [3], also allows permeation of divalent cations. The selectivity filter of nvTRPM2 [33] (Figure 4C) is broader than that of TRPM4 (5.2 Å vs. 4.2 Å in diameter, respectively; Figure 4H, blue vs. pink) [66,72,73], and the pores of human (Figure 4F,G, yellow and orange) and drTRPM2 [59,69] are even slightly wider. However, the selectivity filter is very short, constituted by only three residues, such that the ions entrapped in the structure (modeled as Na^+^; Figure 4D, green spheres) are coordinated by only two adjacent carbonyl groups (Figure 4D). Surprisingly, only nvTRPM2 possesses the acidic side chain (E1037) in the third position of its filter (N997 in drTRPM2, Q981 in hsTRPM2; cf., Figure 1C, left) which had been reported to be absolutely necessary for Ca^2+^ permeation in other TRPM channels [74,75,76]. Nevertheless, all TRPM2 orthologues are Ca^2+^ permeable, despite filter sequences that closely resemble those of Ca^2+^ impermeable TRPM4 and 5 [3,4] channels (Figure 2, blue box). Probably, the short and wider filter permits Ca^2+^ entry through TRPM2 channels, but the acidic side chain in nvTRPM2 further promotes divalent cation permeation [77], as the permeability ratio pCa^2+^/pNa^+^ is much higher for nvTRPM2 compared to hsTRPM2 (~35 vs. ~0.5).

**Figure 3 life-11-00397-f003:**
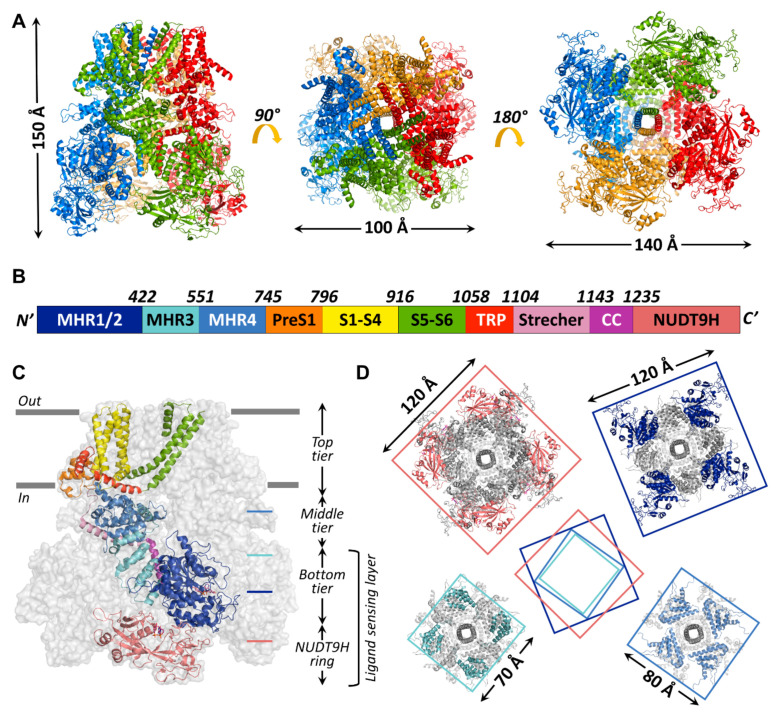
Global architecture of the hsTRPM2 protein. (**A**) Overall structure of hsTRPM2 (6pus) viewed from different angles (side, top, and bottom from left to right) with indicated molecular dimensions. Protomers are colored separately. (**B**) Box representation of the primary sequence segments of a subunit, with segment boundaries indicated. (**C**) Cartoon representation of an hsTRPM2 subunit (color-coded according to (**B**) shown within the gray surface silhouette of the tetramer. Gray bars indicate membrane surfaces. (**D**) Square-shaped cross-sections (at levels indicated in (**C**)) of the tetramer, with dimensions indicated. Center cartoon illustrates rotation of segments relative to each other.

The extracellular mouth of nvTRPM2 contains four negative charges per subunit. The internal vestibule, at the level of the S6 bundle crossing, also contains more negative side chains than TRPM4. Furthermore, both the external and internal vestibules are narrower by almost 50% in nvTRPM2 compared to the TRPM4 channels, leading to a larger negative surface charge density (Figure 4E). Together with a wider pore diameter (Figure 4B,C,H), this confers higher ionic conductance to the channel (~150 pS vs. ~25 pS for nvTRPM2 and TRPM4 channels, respectively, in symmetrical ionic conditions). The lack of some of the negative residues in the vestibules of the hsTRPM2 protein lowers the conductance of its channel pore to a value in between those of nvTRPM2 and TRPM4 (~50 pS). The same negative charges in the external mouth of nvTRPM2 channels are also important for pore stability. Presumably, in the narrow external vestibule, charge repulsion between the closely apposed D1041 and E1042 side chains (Figure 4E) of the post-filter helix (LDE sequence; Figure 1C andFigure 2, green box) prevents pore collapse, allowing nvTRPM2 channels to remain active for a long time in cell-free measurements (Figure 1B, right). In contrast, hsTRPM2 channels in which a GY doublet substitutes for the LDE triplet (Figure 1C andFigure 2, green box) rapidly inactivate (Figure 1B, left). Reverting the sequence to LDE in hsTRPM2 prevents channel rundown (also see earlier discussion) [36], whereas converting the post-filter sequence to human-like in nvTRPM2 introduces channel inactivation [33]. Alignment of post-filter helix sequences revealed that the loss of conserved acidic residues first appeared in vertebrates during evolution (Figure 1C, red vs. blue sequences). Invertebrate channels bear one or two conserved aspartates or glutamates in this region and those that have been studied show no sign of rundown in inside-out patches, in contrast to the inactivation, with variable kinetics, observed for all vertebrate TRPM2 channels studied so far [34].

In ligand-free or mono-liganded structures (6co7, 6bcj, 6mix, 6puo), the main constriction of the pore is the S6 bundle crossing with a radius of ~1 to 2 Å (Figure 4B,C), which precludes ion flux. Thus, these models all represent closed channel conformations. The constricting residue has a bulky side chain, but its chemical nature is variable among orthologues (Ile, Gln, Asn), suggesting that block of ion permeation is simply due to physical obstruction of the tunnel. Such a gating mechanism is widespread among K^+^ channels. Ligand binding to cytosolic domains induces conformational changes in channel structures, which are allosterically translated into a dilation of the S6 gate. Capturing an open state in a cryo-EM structure has proven extremely challenging, as reflected by the fact that out of the 14 TRPM2 structures—and indeed, out of the more than 120 available TRP channel structures [78]—only one drTRPM2 structure (6drj, [59]) adopts the open conformation.

The presence of Ca^2+^ alone has no major effect on the architecture of the TMD, and exerts only subtle effects on the global structure: the structure of hsTRPM2 in EDTA (6puo, 6mix [58,69]) is very similar to that of nvTRPM2 with Ca^2+^ bound at its binding site (6co7, [33]). In contrast, ADPR binding exerts major conformational changes. Already when present alone, it introduces substantial rearrangements in the cytosolic domains (dubbed “priming”, see later discussion). The simultaneous presence of ADPR and Ca^2+^ led to pore opening in drTRPM2, while it induced markedly different structural rearrangements in the two available hsTRPM2 structures. Ca^2+^ addition to ADPR-bound hsTRPM2 was without major structural effects in one study (6pus vs. 6pur, [58]), whereas in another study it induced partial gate opening (6mj2 vs. 6miz, [69]).

In one of the hsTRPM2 structures (6pus, [58]), a ~5° clockwise (when viewed from the cytosol) rigid body rotation of the VSLDs around the central axis was not accompanied by a similar rotation of the S5 and S6 segments; therefore, the bundle-crossing gate was not dilating, and the channel pore remained closed despite large remodeling in the cytosolic parts. The lack of pore opening might have been due to the loss, during sample preparation, of a disulfide bond in the extracellular loop region. This disulfide bond, observed in both the apo and open states of drTRPM2 (Cys1012-Cys1024), is important for the stabilization of the pore, as mutation of these cysteines leads to loss of channel function in whole cells [79]. Indeed, density for the filter and the post-filter region is completely missing in this structure. In the other hsTRPM2 structure [69], the overall resolution is 6.4 Å, and the TMD is less well resolved. In the fully liganded channel (6mj2), dilation of the S6 gate at the level of I1045 and Q1053 is observed, but only to a physical diameter of ~4 Å, which allows the passage of dehydrated or partially dehydrated ions but precludes the flux of fully hydrated ions. The pore radius is very similar to that of the selectivity filter. Such partial opening of the gate is due to an unwinding of the junction between the S6 and TRP H1 segment, which forms a single continuous helix in the apo and ADPR-bound states. The emerging loop might allow more conformational flexibility to the distal parts of S6, and also forms a new contact with the adjacent S6 helix. Such remodeling might be important for the outward rotation, away from the central axis, of the side chains that form the restriction of the gate.

Gating conformational changes are nicely resolved in the drTRPM2 structures [59]. The TRP H1 helix is stacked between MHR4 helices from the bottom and the S4–S5 linker helix from the top (6drk). Upon simultaneous binding of ADPR and Ca^2+^ (6drj) the MHR4 layer moves towards the membrane (see later discussion) and pushes the TRP helix against the S4–S5 segment, which flips across the TRP helix from one side (closer to the central axis) to the other. Such swapping of the linker moves S5, and with it S6, in the extracellular direction by half a helical turn. This allows outward bending of the distal parts of S6, removal of the N1064 and Q1068 side chains from the permeation pathway, and thereby opening of the inner gate. Dilation is vast, producing a tunnel with a diameter of >8 Å, allowing even the flow of hydrated Ca^2+^ ions. Notably, in this structure, the selectivity filter diameter (~5.2 Å) becomes limiting for the permeation of hydrated Ca^2+^ ions. Hence, either this structure still represents a pre-open conformation, or the passage of ions through the filter occurs in a partially or fully dehydrated form. Unfortunately, the currently available structures do not have enough resolution in the pore region to demonstrate ion coordination with certainty (cf [71]). The role of flipping between TRP H1 and the S4-S5 linker seems crucial for gate opening. TRP H1 participates in Ca^2+^ coordination; therefore, Ca^2+^ binding to its binding site might facilitate movement of the TRP helix. Although little priming effect by Ca^2+^ alone was observed in structures (see earlier discussion), a repositioning of the S3 helix is visible in the fully liganded vs. apo structures of drTRPM2, which prepares space for the movement of the S4–S5 segment. The flipping of the latter across TRP H1 is also observed, albeit only in two opposing subunits, in a drTRPM2 structure that obeys a two-fold central symmetry [80]. In this structure (6d73), solved in the presence of added Ca^2+^ but perhaps also containing some ADPR entrapped during sample preparation, two opposing subunits adopt a closed-like while the other pairs adopt an open-like conformation, with a flipped S4–S5 segment. The authors suggested that such an intermediate conformation describes a transient state that precedes channel gate opening. Thus, the drTRPM2 (6drj) and hsTRPM2 (6mj2, 6pus) structures may represent fully or pre-open conformations, rather than an inactive state.

### 4.4. The Ca^2+^ and PIP_2_ Binding Sites of the TRPM2 Channel

While it was long anticipated that Ca^2+^ has to bind to an intracellular crevice close to the pore of TRPM2 [30], the actual binding site was first described in TRPM4 among TRPM channels [72]. The location of the binding site and the coordination of the Ca^2+^ ion is highly conserved among TRPM channels, and recently bound Ca^2+^ was resolved in several TRPM2 structures (6co7 [33], 6drj [59], 6mj2 [69], 6pus and 6puu [58], 6pkx and 6d73 [80]). The Ca^2+^ binding site is located in a narrow cavity at the membrane-cytosol interface, surrounded from the side by the cytosolic ends of S2 and S3, and the short S2-S3 helix (Figure 4J, yellow), and from the bottom by the TRP H1 helix. In nvTRPM2, the Ca^2+^ ion is coordinated in a pentacovalent manner by the side chains of a glutamate (E893) and a glutamine (Q896) from S2, plus an asparagine (N918) and an aspartate (D921) from S3 (Figure 4J, sticks) [33]. The side chain of E1110 from TRP H1 forms a polar contact with Q896, but is too far from the cation to directly bond with it. These five positions are highly conserved among the TRPM2, 4, 5 and 8 channels (Figure 2, red boxes), and play important roles in Ca^2+^ binding, although the exact coordination geometry is slightly different among species. In human and zebrafish TRPM2, the TRP helix glutamate (E1073 and E1088, respectively), besides stabilizing the orientation of the Gln on S2, also directly binds the cation. Mutations of these residues to alanines markedly lowered the apparent Ca^2+^ affinity of channels, confirming their role in direct ion coordination [33]. In closed channels, the Ca^2+^ ions can reach their binding sites only from the cytosol, using two access pathways. One pathway is through a “peripheral tunnel” that connects the binding site to the protein surface. In addition, MHR3/4 embraces a large aqueous cavity, which is connected to the cytosol through large lateral fenestrations, allowing rapid ion exchange between the cavity and the cytosol. Ca^2+^ can reach its binding sites also from this cavity, through a "central tunnel". In fact, the binding site can be considered as a midpoint recess along the joined central and peripheral tunnels. As the TRPM2 pore is also permeable to Ca^2+^, influx of the ion through an open pore can feed the binding sites with their ligand even in situations when cytosolic Ca^2+^ levels are low [30]. This phenomenon was proposed to have physiological relevance. In the presence of ADPR alone, TRPM2 channels open very rarely, but a low spontaneous activity is observed. Once a channel opens, extracellular Ca^2+^ floods its cytosolic cavity from where it titrates the Ca^2+^-binding sites, increasing channel open probability. Upon pore closure, the extracellular Ca^2+^ supply is cut; however, the pore is likely to re-open before all the Ca^2+^ has dissipated from the cytosolic crevices, unless cytosolic Ca^2+^ is rapidly and completely washed out [30,33]. The structure–function studies therefore suggest that in living cells an initial igniting puff of Ca^2+^ might be sufficient to open ADPR-bound TRPM2 channels, which will maintain a steady influx of cations as long as ADPR is available. Such a mechanism is in sharp contrast to TRPM4 or 5 channels, which are also Ca^2+^-activated, but Ca^2+^ impermeable. In these channels, Ca^2+^ can access their binding sites only from the cytosol, requiring an intracellular Ca^2+^ source.

Truncation of side chains that form the Ca^2+^ binding site not only caused rightward shifts in the Ca^2+^ dose-response curve of nvTRPM2 channels, but also severely reduced maximal channel open probability. The addition of exogenous PIP_2_ potently stimulated currents of such mutant channels in saturating Ca^2+^ conditions [33]. Conversely, depletion of free PIP_2_ in inside-out patches reduced hsTRPM2 currents and suppression could be reversed only by millimolar amounts of Ca^2+^ [36]. While these data might be explained by the additive effects of Ca^2+^ and PIP_2_ on the stability of the open conformation, they also raise the possibility that binding of PIP_2_ affects Ca^2+^ coordination. PIP_2_ is a well-known agonist of TRPM channels [81]. Binding of its head group to some positively charged side chains on the TRPM8 TRP H1 helix stimulates these channels, while Ca^2+^, entering through the open TRPM8 pore, activates phospholipase C, leading to PIP_2_ cleavage and consequent channel desensitization [82]. Basic residues of TRP H1 are also reported to play a role in PIP_2_ binding to TRPM5 and TRPV5 channels [82]. In the nvTRPM2 channel structure, a density for a phosphatidic acid is clearly visible near the TRP H1 helix; however, the headgroup density could not be resolved (Figure 4J, top, salmon sticks) [33]. A PIP_2_ molecule was seen in the corresponding cavity, the so-called vanilloid binding pocket of TRV1 channels (5irz, [83]). The acyl chains of PIP_2_ run along the S4 helix of one subunit and the S5 and S6 helices of the adjacent subunit. The inositol headgroup fits into a cleft surrounded by the cytosolic end of S3, the S4–S5 linker, and TRP H1 from the bottom. The PIP_2_ phosphates are mostly coordinated by basic amino acid side chains. While an inositol headgroup can be easily accommodated by the same cleft in the nvTRPM2 apo structure (Figure 4J, top), the side chains of the canonical basic residues of the TRP box sequence (VWKFQR) point away from the lipid. However, possibly, ligand binding might rotate the TRP helix along its central axis, leading to better coordination. If that is the case, then PIP_2_ binding at this site could, on the one hand, directly affect Ca^2+^ coordination through an interaction with S3, and on the other hand, through interactions with TRP H1, together with Ca^2+^, promote conformational changes at the S6 bundle crossing that trigger pore dilation. PIP_2_ binding at the vanilloid pocket was also observed in the TRPV5 [84] and the KCNQ1 K^+^ channel [85]. Similarly to TRPV1, the latter structures were solved in nanodiscs, which helps to resolve native membrane lipids. While phosphatidic acid (thus, potentially PIP_2_) was resolved at the vanilloid site in other TRP channel structures (5z96, mouse TRPC4 channel [86]; 5vkq, Drosophila NOMPC channel, [87]), this is not the exclusive location of lipid binding. In the TRPM8 structure (6nr2 and 6nr3), PIP_2_ binds to the opposite side of the VSLD and fits into a pocket formed from the side by the preS1, S1, and S4 helices, from the bottom by TRP H1 and MHR4 helices, and from the other side by the adjacent S5 segment [88]. A phosphatidic acid density is also resolved in the corresponding cleft in both the NOMPC [87] and the nvTRPM2 (Figure 4J, bottom, salmon sticks) [33] structures. Currently, more nanodisc-based structures and functional studies are needed to clearly resolve the PIP_2_ binding site(s) in different TRPM channels.

### 4.5. The ADPR Binding Sites and Molecular Gating of the TRPM2 Channel

Based on its sequence homology with soluble ADPR hydrolases, the NUDT9-H domain of TRPM2 had long been anticipated as the primary ligand binding site for channel activation; however, the presence of a ligand at this site was not observed until recently [58]. In hsTRPM2 ADPR is clamped in a cleft between the NUDT9-H cap subdomain and a β-strand region of the core subdomain (Figure 4I, lower inset), similarly to ribose-5-phosphate bound to mitochondrial NUDT9 [22]. The adenine ring intercalates between the side chains of D1431 and Y1485, while the alpha-phosphate is coordinated by R1433. Truncation of the R1433 side chain or deletion of the full NUDT9-H domain abolished hsTRPM2 currents in inside-out patches, attesting to the importance of ADPR binding to NUDT9-H for channel activity. Similarly to human channels, deletion of the NUDT9-H domain also abolished currents of drTRPM2 [59]. Surprisingly, however, a similar truncation preserved channel function of nvTRPM2, which strongly suggested the presence of a second ligand binding site [56]. Indeed, in recent agonist-bound structures of drTRPM2 and hsTRPM2 (6drj and 6pus), a second ADPR molecule is observed, bound in a pocket formed by residues of MHR1/2 (Figure 4I, top inset) [58,59]. ADPR is bent into a U-shape, with adenine and terminal ribose rings approaching each other, in contrast to the elongated conformation of the ligand bound at the NUDT9-H domain (Figure 4I, lower inset). In hsTRPM2, the adenine ring stacks against the side chain of Y295; the phosphates and the terminal ribose are coordinated by two Arg residues, R358 and R302, respectively. These coordinating residues are highly conserved in all TRPM2 orthologues (Figure 2, orange boxes), suggesting that the MHR1/2 site is an ADPR-binding site preserved by evolution. Alanine mutation of the tyrosine and combined truncation of the two arginines markedly lowered or abolished channel currents in human and zebrafish TRPM2, respectively [58,59], and individual truncations of the three side chains also impaired nvTRPM2 gating [57]. Thus, the MHR1/2 site seems to play a major role in ADPR-dependent TRPM2 channel gating in all orthologues. Of note, in one study, the same double arginine truncation of hsTRPM2 did not eliminate Ca^2+^ uptake in cells exposed to hydrogen-peroxide [69], leading the authors to conclude that the MHR1/2 site does not play a major role in hsTRPM2. Cross-species comparison of available structures suggests that the general assembly of channel orthologues is similar but not identical. A major variable is the relative contribution of the NUDT9-H domain to stabilization of the MHR1-4 domains. The NUDT9-H domain is not resolved in the otherwise high-resolution structure of nvTRPM2 [33], suggesting that it is loosely connected to the MHR1-4 domains. In human channels the NUDT9-H domain is engaged in three interfaces that stabilize the MHR regions: two intrasubunit interfaces between MHR1/2 and MHR3, and an intersubunit interface with a neighboring MHR1/2. The latter interaction hub undergoes a major rearrangement upon ADPR binding, but is absent in drTRPM2 [59], possibly due to a sequence deletion of the P-loop region as compared to other orthologues [69] (Figure 2, gray bar). Correspondingly, in hsTRPM2, NUDT9-H acts as a wedge that glues together adjacent MHR1/2 domains and also fixes the MHR1/2 ring to the MHR3 regions in the apo state, while in apo-drTRPM2, it is dislocated from the cleft between adjacent MHR1/2 domains and is more loosely attached to the MHR1/2 ring. The functional consequences of such a different structural organization on the activation of human vs. zebrafish channels are unclear at the moment. In human channels, the strong interaction between MHR1/2 and MHR3 needs to be broken to allow conformational changes triggered by ADPR binding at the MHR1/2 site to be propagated through MHR3/4 to induce channel opening. It has been suggested that this process might be initiated by ADPR binding to the NUDT9-H domain. As the ligand inserts into its binding cleft, it induces an ~3° clamshell-like closure of the core and cap subdomains around a hinge in the β-strand region. Tightening of NUDT9-H triggers its disengagement from the resting state of the intersubunit interface, yielding a more flexible conformation, similar to that already present in unliganded drTRPM2 [58]. Further findings seem to support such different activation mechanisms for human and drTRPM2 channels. First, in the drTRPM2 structures, the presence of ADPR causes no significant remodeling in the NUDT9-H domain, and no density for a bound ligand is detected [59]. Second, microscale thermophoresis experiments on isolated NUDT9-H domains revealed a markedly lower ADPR affinity for drNUDT9-H vs. hsNUDT9-H (K_d_ ~1 mM vs. ~40 μM, respectively [69]). Of note, various estimates of the K_d_ value of the isolated hsNUDT9-H domain for ADPR fall into the tens-of-micromolar range (15–130 μM; [24,69]), similarly to the K_m_ value for ADPRase activity of isolated nvNUDT9-H or full-length nvTRPM2 (8-18 μM) [34]. In contrast, for both channel orthologues, the EC_50_ for channel activation by ADPR is 1 to 2 μM [30,33]. All in all, these findings have led to the suggestion that ADPR binding to NUDT9-H is essential to prime hsTRPM2 channels, whereas ligand binding exclusively at the MHR1/2 site is sufficient for activation of drTRPM2 [58]. If so, it is still unclear why deletion of the C-terminal site abolishes currents also for drTRPM2 channels.

ADPR binding to its MHR1/2 site induces a clamshell-like closure of the domain and a corresponding remodeling of the entire MHR region in hsTRPM2. While the MHR1/2 ring expands and moves closer to the TMD, the NUDT9-H ring contracts towards the coiled coil and engages in a new activation interface with MHR2 [58]. Subsequent conformational changes are similar in human and zebrafish TRPM2. The MHR3/4 region plays a key role in translating movements of the MHR1/2 domain into a rotation of the stretcher helices and dislocation of the TRP helices. The latter are directly connected to the S6 segments; hence, they finally signal to the TMD and initiate pore dilation (see earlier discussion on the TMD). Snapshots of a potential intermediate state of the opening conformational transition have been published, reporting a two-fold central symmetry for a Ca^2+^-bound TRPM2 structure [80] in which weak unassigned density in the MHR1/2 site might have originated from endogenous ADPR. The density for Ca^2+^ and ADPR seemed stronger in two opposing subunits, which adopted an open-like conformation, but weaker in the other two protomers that adopted a closed-like conformation. While such intermediate states were also published for other channels [89,90,91,92,93] including TRP channels [94,95] and might potentially reflect physiologically relevant ligand-induced structural rearrangements, more structural snapshots will yet be required to fully understand gating conformational transitions of TRPM2.

The inhibitory ligand 8-Br-cADPR binds only to the MHR1/2 site of hsTRPM2, and no density is observed at the NUDT9-H domain [58]. 8-Br-cADPR is a cyclic molecule whose shape resembles the U-shape conformation of ADPR bound in the MHR1/2 site (Figure 4I, top inset); however, its bromide group clashes with Y285; hence, it cannot fully insert into the ligand binding cleft. The conformation of the channel resembles that of unliganded hsTRPM2, as expected for an antagonist-bound protein.

In vertebrate channels, the NUDT9H domain is catalytically inactive (Figure 1D, center) [24,35,69] and both MHR1/2 and NUDT9-H ADPR-binding sites seem to play a role in channel activation [58,59,69], which precludes the experimental profiling of individual nucleotide-binding properties of the two sites. In nvTRPM2, however, channel gating is independent of NUDT9-H [56,57], whereas the ADPRase activity of NUDT9H is independent of the rest of the protein [57] (Figure 1E). This allows experimental dissection of the ligand binding properties of the MHR1/2 site (reported by EC_50_ for channel gating) and the NUDT9-H site (reported by the K_m_ of enzymatic activity). Nucleotide preference of the gating site is markedly different in human vs. *Nematostella* TRPM2 channels [57,96]. As an example, ADPRP is a partial agonist of hsTRPM2, but the most potent stimulator of nvTRPM2 [57]. Furthermore, in nvTRPM2, the nucleotide preferences of the N- and C-terminal sites also differ from each other. Inosine diphosphate-ribose (IDPR) was reported to preferentially act on the NUDT9-H domain rather than the N-terminal site of nvTRPM2, acting as a competitive inhibitor of its ADPR hydrolase activity [96]. Quantitative reconstruction of the affinity series of a set of ADPR analogs for the two distinct sites in nvTRPM2 revealed that nvNUDT9-H is not very selective among the tested nucleotides, while the MHR1/2 activation site is highly sensitive to the nature of the ligand [57]. Alanine mutations of residues, involved in ADPR coordination at the MHR1/2 site in vertebrate TRPM2 channels, strongly affected the EC_50_ values for nvTRPM2 channel activation by both ADPR and ADPRP, but not by Ca^2+^. This suggests that the truncations indeed affected ADPR binding at the "gating site", and that this site in nvTRPM2 corresponds to the MHR1/2 binding site in vertebrate channels. EC_50_ values are affected by the K_d_ of a ligand in both the closed- and the open-channel state (K_d,c_ and K_d,o_, respectively), which are not identical: a stimulatory ligand binds more tightly in the open than in the closed conformation. The ratio K_d,o_/K_d,c_ represents the efficacy of the agonist. Therefore, mutating a residue that coordinates the ligand with distinct interaction energies in closed vs. open states will differentially affect K_d,c_ and K_d,o_, and consequently alter not only EC_50_, but also the efficacy of the ligand. Hence, analysis of the maximum open probabilities (i.e., the open probability of fully liganded channels in the presence of a saturating agonist (ADPR or ADPRP) concentration) and EC_50_ values of WT and mutant nvTRPM2 channels allows functional mapping of the MHR1/2 site. While the side chains of T143 and T307 (corresponding to T150 and T312 in drTRPM2, respectively) seem to coordinate the β-phosphate and the terminal ribose in a statical manner, the strength of the interactions between the side chains of T179, R329 and Y264 (corresponding to T186, R334 and Y271 in drTRPM2, respectively) and the ADPR adenine ring and α-phosphate change during ADPR-induced conformational remodeling [57]. The role of R271 (corresponding to R278 in drTRPM2), a residue near the terminal ribose of ADPR, is not yet fully understood; potentially, it might antagonize ADPR binding in the closed state.

## 5. Conclusions

In 2017, the Nobel Prize in Chemistry was awarded “for developing cryo-electron microscopy for the high-resolution structure determination of biomolecules in solution”. It honored the laureates for their long-lasting efforts over decades to develop and improve the method. Yet, it was not until recently that single-particle cryo-EM has attained wide-spread use thanks to advanced equipment, in particular, high acceleration voltage microscopes with single-electron detectors, and also advanced software packages. As a consequence, an unprecedented output in novel structures is now available. In the last decade, almost 7000 structures solved by electron microscopy have been deposited, including several structures of the TRPM2 ion channel. The protein is a homotetramer, with large cytosolic regions, mostly constituted by the N-terminal sequences. The TMDs contain the cation permissive pore which allows the flux of both mono- and divalent cations. Such a non-selective pore is mostly explained by a wide and short selectivity filter. The TMD shares homology with that of voltage-gated K^+^ channels; however, due to the loss of conserved positive residues in the S4 helix, TRPM2 is largely voltage insensitive. The channel is activated by cytosolic Ca^2+^, ADPR and PIP_2_. The Ca^2+^ binding site is located at the membrane-cytosol interface in a deep crevice surrounded by S2 and S3 within the VSLD. Ca^2+^ can reach its ligand binding site not only from the bulk solution but also from the internal cavity embraced by the cytosolic domains. This aqueous cavity is directly connected to the channel pore; therefore, Ca^2+^ entering through the open pore can rapidly saturate its binding sites and keep channels open, explaining the slowing of channel closure at high extracellular Ca^2+^ levels. The channels contain two ADPR-binding sites. One is embedded in MHR1/2, the other is formed by the C-terminal NUDT9-H domain. While the MHR1/2 site is the principal ligand binding site required for channel activation in all species, the NUDT9-H domain is dispensable in invertebrate, but not in vertebrate channels. In the latter channels, the NUDT9-H domain likely supports remodeling of the cytosolic domains during gating transitions. In invertebrate channels, NUDT9-H is linked to the channel in a flexible manner and does not interact with MHR1-4. On the other hand, it hydrolyses the bound ADPR to release AMP and ribose-5-phosphate. In the course of evolution, this hydrolytic activity of NUDT9-H was lost in vertebrates, in parallel to the appearance of pore inactivation. It is tempting to speculate that in invertebrate cells ADPR hydrolysis serves as a major regulatory mechanism by clearing away the activating ligand, while vertebrate cells limit the open time of their TRPM2 channels through pore inactivation, instead of ligand hydrolysis.

Single particle cryo-EM brought remarkable advances in understanding the structure–function aspects of TRPM2 channels, but several questions are still open. PIP_2_ is an obligate activator of the channel, but its exact binding site is still elusive. More structural models of nanodisc-reconstituted proteins might be needed to identify that binding site. Additionally, while ADPR binding at the MHR1/2 site is fundamental for channel activation, the fine prints of species-specific mechanisms of activation and the relative contributions of the MHR1/2 vs. NUDT9-H sites await further clarification.

## Figures and Tables

**Figure 1 life-11-00397-f001:**
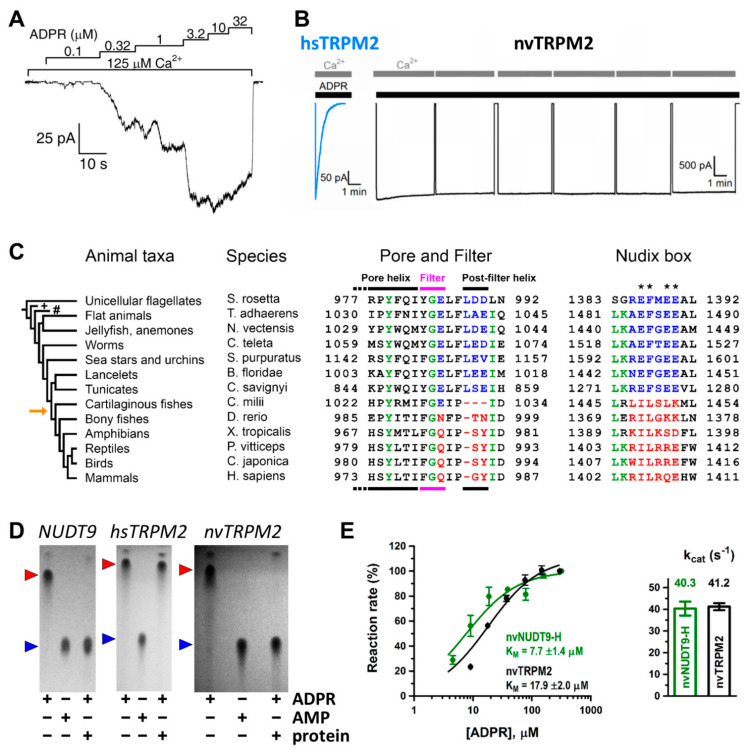
Functional characterization of TRPM2. (**A**) Representative trace of a macroscopic inside-out patch-clamp measurement. Downward deflections reflect channel opening and flux of Na^+^ ions in response to indicated stimulating ADPR and Ca^2+^ concentrations. Scaling of current and time are indicated (L-bar). (**B**) hsTRPM2 currents inactivate quickly in the continuous presence of ADPR and Ca^2+^ in contrast to long persisting currents of nvTRPM2. Intermittent Ca^2+^ removals help identify the baseline. (**C**) Alignment of pore and Nudix sequences among animal species. The taxa Porifera (sponges; marked with ’+’) and Ctenophora (comb jellies; marked with ’#’) did not return any TRPM2-like sequences in BLAST. Orange arrow next to evolutionary tree shows appearance of vertebrae. Canonical sequence motifs of invertebrate channels are highlighted in blue; vertebrate substitutions are indicated in red. Asterisks indicate Nudix box residues that are important for ADPR hydrolysis. (**D**) Thin layer chromatography reveals hydrolysis of ADPR (red triangle) (as formation of AMP; blue triangle) by mitochondrial NUDT9 enzyme and by the isolated NUDT9-H domain of nvTRPM2, but lack of ADPR hydrolysis by the hsNUDT9-H domain. (**E**) ADPR dose-dependence of hydrolysis and molecular turnover number (k_cat_) for full-length nvTRPM2 protein (black) and for the isolated nvNUDT9-H domain (green). Panel A was adapted from [31], panel B was adapted from [33], panels C and E were adapted from [34], panel D was adapted from [35] (left and middle) and [34] (right).

**Figure 2 life-11-00397-f002:**
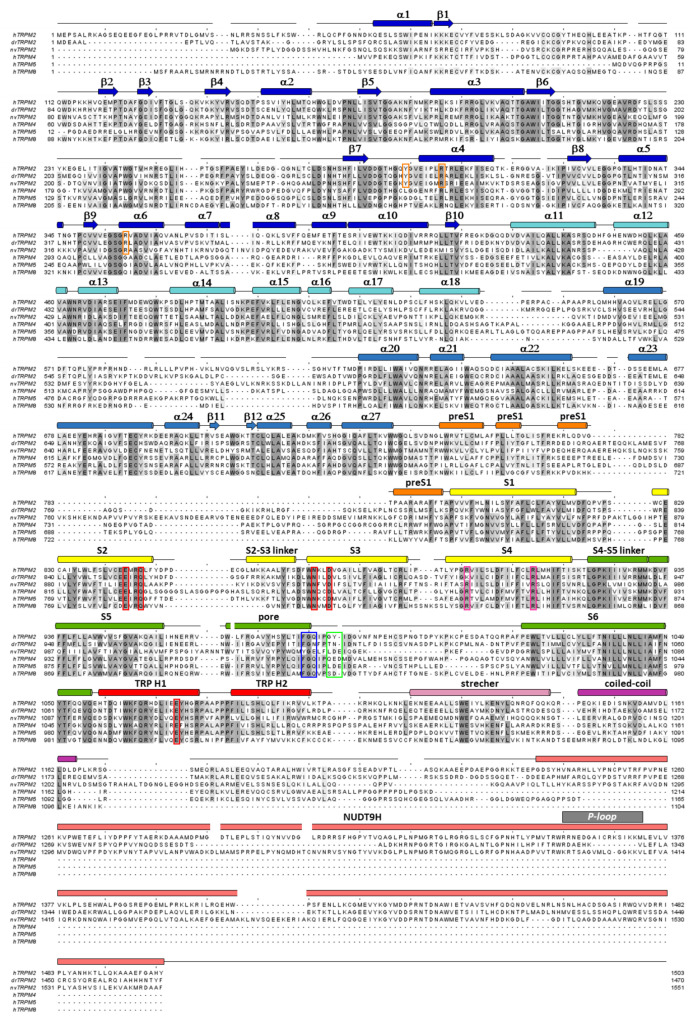
Multiple sequence alignment of TRPM proteins. Sequence alignment of TRPM2, TRPM4, TRPM5 and TRPM8 channel orthologues (hs: *Homo sapiens*, dr: *Danio rerio*, nv: *Nematostella vectensis*) calculated using Clustal Omega [54], with some manual adjustments. Conserved residues are highlighted according to percent of identity in Jalview [55]. Secondary structure elements (based on 6pus) are shown for hsTRPM2 above the sequences. Color coding follows that used in Figure 3B. Canonical Arg residues in the S4 segment (magenta) and residues of the selectivity filter (blue), the post-filter helix (green), and of the ADPR1 (orange) and Ca^2+^ binding sites (red) are highlighted in boxes. P-loop segment of the NUDT9-H domain is identified by gray bar.

**Figure 4 life-11-00397-f004:**
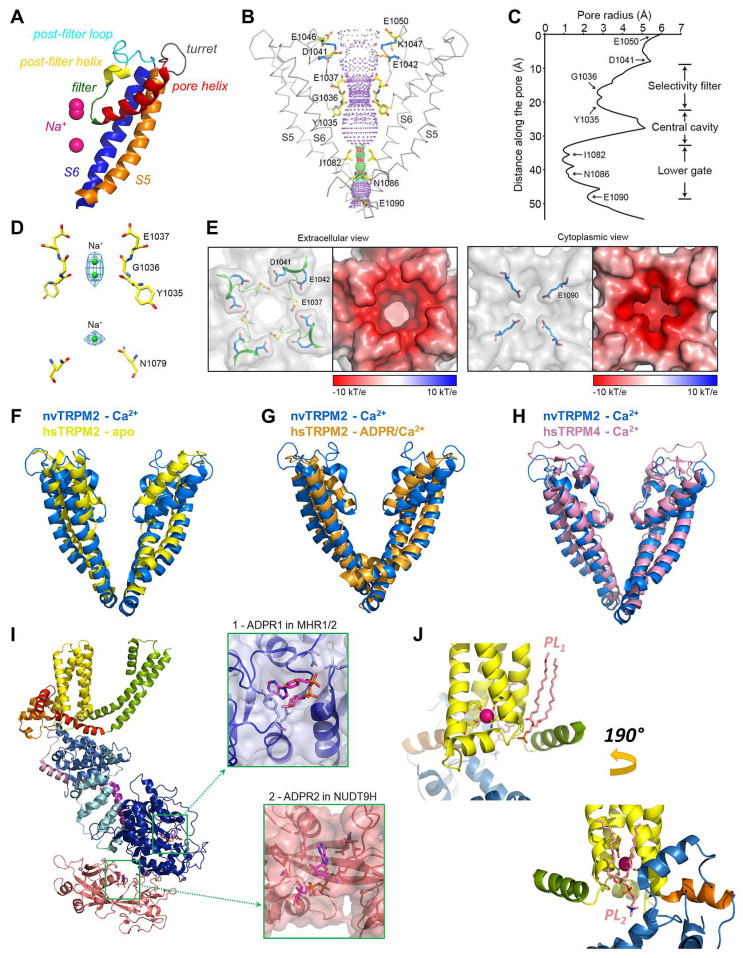
The pore and ligand-binding sites of TRPM2. (**A**) Cartoon representation of pore-domain segments of nvTRPM2 with modeled Na^+^ ions (purple spheres) (6co7, (**A**–**E)**). (**B**) Internal surface of the pore (purple dotted mesh) indicating restrictions (red, radius < 1.15 Å and green, radius 1.15 to 2.30 Å). Pore-lining residues are shown in stick representation with elementary color coding (The PyMOL Molecular Graphics System, Version 2.0 Schrödinger, LLC.). (**C**) Van der Waals radius of the pore along its central axis. (**D**) Na^+^ ions (green spheres) modelled into densities (blue mesh) observed in the selectivity filter and the central cavity. (**E**) Surface electrostatics of the outer (left panel) and inner pore (right panel) vestibules. Left insets: pore residues are highlighted within surface representation; right insets: surface electrostatics, negative (red) to positive (blue), calculated at pH 7. (**F**–**H**) Superpositions of the pore region of nvTRPM2 (6co7, blue) with those of apo hsTRPM2 (6puo, yellow), ADPR/Ca^2+^-bound hsTRPM2 (6pus, orange), and Ca^2+^-bound hsTRPM4 (6bqv, pink). (**I**) Protomer of hsTRPM2 (6pus), color-coded as in Figure 3C. Insets show expanded views of the ADPR1 and ADPR2 ligand binding sites (MHR1/2 and NUDT9-H, respectively). (**J**) Phospholipid (PL, salmon sticks) binding to the vanilloid-type (top panel) and the preS1-VSLD-TRPH site (bottom panel) in nvTRPM2 (6co7). Sticks are shown for residues on S2 and S3 that coordinate the Ca^2+^ ion (magenta sphere). Panels (**B**–**E**) were adapted from [33].

## Data Availability

Not applicable.

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
