# Peer review of "Two Decades of Evolution of Our Understanding of the Transient Receptor Potential Melastatin 2 (TRPM2) Cation Channel"

_life, 2021, doi:10.3390/life11050397_

Round 1

Reviewer 1 Report

This review written by A. Szollosi is a particulalry interesting paper, summarizing the results of the major TRPM2 ion channel-related researches. The review is nicely and logically written. The quality of the figures is fine. The author precisely collected and reviewed the literature data during the work. In my opinion, the author fully explained all aspects of the structure and molecular functions of the TRPM2 channel, therefore I suggest to accept this manuscript in its present form.

Reviewer 2 Report

This is a technically detailed review of the TRPM2 cation channel. The manuscript provides a useful summary of recent insights on the structure and function/regulation of the channel and should serve as an informative reference in the field.

Comments:

1) Please read carefully to ensure sentence structure and syntax/wording clearly articulate the intended concept/meaning. 

Examples:

Line 185, ....and ribose-5-phosphate are also no TRPM2 channel activators.

Line 42,...TRPM2 activity is 'inevitable' for many physiological processes...

The word 'embrace' is used repeatedly, but does not seem to convey a clear message.

2) The large section (4.1) dedicated to single-particle cryo-EM seems a bit out of place...seems the manuscript should focus on the channel or the method and not both. Could focus the work by condensing this section.

3) Fig 4E, print is too small

Reviewer 3 Report

In this manuscript the author summerizied the main structural and biophysical properties of the TRPM2 channel. The manuscript is well written, sections about the activators, the structural properties are well detailed and well suported with the recent literature.
